# Pyrazinamide resistance in rifampicin discordant tuberculosis

**Nomonde Ritta Mvelase**[1,2]*, **Ravesh Singh**[1,2], **Khine Swe Swe-Han**[1,2], **Koleka Patience Mlisana**[1,2,3]

**1** Department of Medical Microbiology, National Health Laboratory Service, Inkosi Albert Luthuli Hospital, Durban, South Africa, **2** School of Laboratory Medicine and Medical Sciences, Department of Medical Microbiology, University of KwaZulu-Natal, College of Health Sciences, Durban, South Africa, **3** Centre for the AIDS Programme of Research in South Africa (CAPRISA), University of KwaZulu-Natal, Durban, Durban, South Africa

* dlaminin15@ukzn.ac.za

## Abstract

### Introduction

*Mycobacterium tuberculosis* strains with phenotypically susceptible *rpoB* mutations (rifampicin discordant) have emerged following implementation of rapid molecular drug resistance testing for tuberculosis. Whilst rifampicin resistance is known to be associated with resistance to other rifamycins (rifapentine and rifabutin) as well as isoniazid and pyrazinamide, rifampicin discordant strains have shown high rates of susceptibility to isoniazid and rifabutin. However, pyrazinamide susceptibly testing results have not been reported.

### Materials and methods

We evaluated pyrazinamide resistance in 80 rifampicin discordant and 25 rifampicin and isoniazid susceptible isolates from KwaZulu-Natal in South Africa using Mycobacteria Growth Indicator Tube method and sequencing of the *pncA*. We also compared susceptibility of pyrazinamide with that of isoniazid.

### Results

Pyrazinamide resistance was found in 6/80 (7.5%) rifampicin discordant isolates. All pyrazinamide resistant isolates were also resistant to isoniazid and pyrazinamide resistance was found to be associated with isoniazid resistance. No pyrazinamide resistance was found among the isoniazid susceptible isolates.

### Conclusion

Given the low prevalence of pyrazinamide resistance in rifampicin discordant TB, this anti-TB drug still has a significant role in the treatment of these patients. Performing pyrazinamide susceptibility testing remains a challenge, our findings show that isoniazid susceptible isolates are unlikely to be resistant to pyrazinamide among the discordant TB isolates.

---

**Data Availability Statement:** All relevant data are within the paper.

**Funding:** NRM received funding from the University of KwaZulu-Natal for this study. The funder had no role in study design, data collection

and analysis, decision to publish, or preparation of the manuscript.

**Competing interests:** The authors have declared that no competing interests exist.

## Introduction

The implementation of rapid molecular drug resistance testing for tuberculosis (TB) has greatly improved the diagnosis of drug resistant TB (DR-TB). Although molecular tests have replaced phenotypic drug susceptibility testing (DST) in the initial diagnosis of DR-TB, the latter remains the gold standard. Consequently, soon after the implementation of molecular tests, reports began to emerge showing *M. tuberculosis* isolates with *rpoB* mutations that were phenotypically susceptible to rifampicin (herein referred to as rifampicin discordant) [1,2]. Later, it became apparent that rifampicin discordant isolates were not uncommon, accounting for 10–20% of *M. tuberculosis* isolates with *rpoB* mutations [3,4].

The specific mutations responsible for this rifampicin discordance have been identified and proven to be different to those responsible for high level rifampicin resistance (rifampicin concordant). The most commonly described rifampicin discordant mutations include; L430P, D435Y, H445L, H445N and L452P while the most common concordant mutations include S450L, D435V, H445Y and H445D [5–7]. In addition, while it is well known that rifampicin resistance is associated with resistance to other rifamycins (rifabutin and rifapentin) as well as isoniazid and pyrazinamide [5,8–10], recent studies have shown that the majority of rifampicin discordant isolates remain susceptible to rifabutin and a significant amount is still susceptible to isoniazid [11,12]. However, susceptibility of rifampicin discordant isolates against pyrazinamide has not been studied.

Pyrazinamide is a critical drug in the treatment of both drug susceptible and DR-TB. Although the exact mechanism of action is not known, it requires activation to an active form called pyrazinoic acid [13,14]. This activation is performed by the pyrazinamidase enzyme encoded by the *pncA* gene. Consequently, mutations in the *pncA* gene are the main cause of resistance against pyrazinamide, accounting for about 90% of resistance [15].

Pyrazinamide is unique to other TB drugs because it only acts under acidic conditions and it targets the slowly multiplying *Mycobacteria* that persist in these environments including those located inside the macrophages [16]. It was this unique activity that led to the reduction of duration of therapy for drug susceptible TB from nine months to six months. However, the need for an acidic environment has complicated the in vitro susceptibility testing as acidic conditions that are required for testing are also detrimental to *Mycobacteria* [17]. As a result, pyrazinamide DST is not routinely performed. The rifamycins, isoniazid and pyrazinamide are the crucial drugs in the standard first line TB treatment regimen.

Despite rifampicin discordant *M. tuberculosis* being a distinct entity from rifampicin resistant (concordant) TB as evidenced by different *rpoB* mutations and antimicrobial susceptibility profile, these patients continue to be managed the same way. This is due to lack of evidence on alternative treatment regimens for rifampicin discordant TB. The second-line treatment regimen is known to be inferior, longer and more toxic compared to the first line treatment. In this study we examined the susceptibility of rifampicin discordant *M. tuberculosis* isolates against pyrazinamide in order to determine the appropriate treatment for the affected patients. In addition, we compared the susceptibility of pyrazinamide to that of isoniazid.

## Materials and methods

### Study design and setting

The study was performed at a central academic South African National Accreditation System (SANAS) accredited TB laboratory of the KwaZulu-Natal (KZN) province in South Africa. The laboratory performs all the *M. tuberculosis* cultures plus both phenotypic and genotypic

drug susceptibility testing for routine clinical work of the public sector of the KZN province. Routine *M. tuberculosis* culture was performed using Mycobacteria Growth Indicator Tube (MGIT, Becton Dickinson) while 1% agar proportion method on Middlebrook 7H10 was used for rifampicin, isoniazid, ofloxacin and kanamycin phenotypic DST at a critical concentration of 1μg/mL, 0.2μg/mL, 2μg/mL and 5μg/mL respectively. Genotypic DST for first line anti-TB drugs was done on MGIT positive specimens using the MTBDR*plus* version 2 assay (Hain Lifescience, Nehren, Germany).

## *M. tuberculosis* isolates

The isolates used in this study were described in our previous report [12]. These were found to be resistant to rifampicin on the Genotype MTBDRplus while susceptible to rifampicin on the phenotypic 1% agar proportion method. In the previous study, 83 rifampicin borderline resistant isolates were investigated. Additionally, 40 time matched susceptible strains (susceptible to the routinely tested anti-TB drugs; isoniazid, rifampicin, ofloxacin and kanamycin) were also examined. In the current study, we performed pyrazinamide susceptibility testing on 80 (of the previous 83) rifampicin borderline resistant isolates as well as in 25 (out of the previous 40) using both phenotypic and genotypic methods.

At the time of the study, the rifampicin critical concentration was still set at 1μg/mL. However, in 2021 the world health organization (WHO) lowered the rifampicin critical concentration to 0.5 μg/mL in an attempt to reduce discordance between phenotypic and molecular assays [18]. When reviewing the rifampicin minimum inhibitory concentration (MIC) of the discordant isolates from our previous report against the new critical concentration of 0.5μg/mL, 45.0% (36/80) of the isolates remained rifampicin discordant (MIC $\leq$ 0.5 μg/mL), while the rest would be regarded as resistant to rifampicin (MIC $>$ 0.5 μg/mL).

## Pyrazinamide phenotypic drug susceptibility testing

Phenotypic DST was performed using the MGIT 10% proportion method at a critical concentration of 100 μg/ml as described in the MGIT manual [19]. However, the MGITs were incubated at 37°C and results were read manually daily using the ultraviolet light. Any MGIT that showed fluorescence was considered as positive. Where the MGIT drug containing tube showed growth within 48 hours of the positive growth control, the isolate was considered as resistant. However, when the drug containing tube showed no growth within 48 hours of the growth control tube, the results was read as susceptible [20]. The *M. tuberculosis* H37Ra (ATCC 25177) was used as a control strain.

## Polymerase Chain Reaction (PCR) and sequencing of the *pncA* gene

The deoxyribonucleic acid (DNA) was extracted using the Quick-DNA™ Miniprep Kit. The *pncA* gene was amplified using published primers; forward (5 = GGCGTCATGGACCCTATA 3 =) and reverse (5 = GTGAACAACCCGACCCAG 3 =) [21]. The amplified product yielded a full length *pncA* gene consisting of 561 base pairs (bp), plus 90bp downstream and 30bp upstream. Both positive and negative control were used during the PCR reaction. Sequencing reaction was performed in the ABI 3500 automated DNA sequencer (Applied Biosystems) using the BigDye Terminator v3.1 cycle sequencing kit and forward primers. The sequences were analyzed using the BioEdit sequence alignment editor with *M. tuberculosis* H37Rv as a wild type reference strain.

## Data analysis

The prevalence of pyrazinamide resistance in rifampicin discordant TB was determined by calculating the proportion rifampicin discordant isolates which demonstrated pyrazinamide resistance on genotypic/phenotypic DST method. Fisher's exact test was used to calculate association between pyrazinamide resistance and isoniazid resistance and the level of significance was set at $P < 0.05$.

## Ethical approval

Ethics clearance was obtained from the University of KwaZulu-Natal Biomedical Research Ethics Council (BE267/18). Because this was retrospective analysis of routine laboratory specimens, no individual patient consent was required.

## Results

Among the 80 rifampicin discordant isolates tested for pyrazinamide susceptibility, six (7.5%) were found to be resistant to pyrazinamide (Table 1). Notably, out of these six isolates, only one was among the 36 (2.8%) which remained discordant after applying the new rifampicin critical concentration of 0.5 μg/mL. Of the 80 rifampicin discordant isolates that were tested for pyrazinamide susceptibility, 48 (60%) were susceptible to isoniazid while 32 (40%) were resistant to isoniazid. Importantly, all six pyrazinamide resistant isolates were among those that were also resistant to isoniazid. Overall, pyrazinamide resistance was associated with isoniazid resistance (p = 0.001). All 25 susceptible isolates were also susceptible to pyrazinamide.

Out of the 80 rifampicin discordant isolates, sequencing of the *pncA* gene revealed a polymorphism is five isolates. This included one deletion in nucleotide position 214 of one isolate and four unique nonsynonymous mutations (T47A, H51P, P54S and S104R) in other four isolates, but sequencing failed on one phenotypically resistant isolate (Table 1). As indicated in Table 1, three of these mutations (T47A, H51Pand S104R have been classified as associated with resistance according to the recently released WHO catalogue of mutations in *M. tuberculosis* [22].

Pyrazinamide phenotypic MGIT DST of 80 rifampicin borderline resistant isolates demonstrated resistance in all but one genotypically resistant isolate. This one isolate was difficult to grow and the MGIT DST revealed growth on the drug containing tubes three days before the control tubes on all three repeat occasions. Therefore, the phenotypic test was regarded as unsuccessful (failed). Among the 25 susceptible isolates, no pyrazinamide resistance was detected by both phenotypic MGIT DST as well as sequencing of the *pncA* gene.

**Table 1. Pyrazinamide phenotypic and genotypic drugs susceptibility results.**

| Codon Position | Genotypic DST | PZA MGIT DST | WHO Classification of *pncA* mutation | INH DST |
|---|---|---|---|---|
| T47A | R | R | Assoc w R | R |
| H51P | R | Failed | Assoc w RI | R |
| P54S | R | R | Not classified | R |
| S104R | R | R | Assoc w RI | R |
| 65/72 Deletion | R | R | Not classified | R |
| Failed | Failed | R | Not applicable | R |

DST: Drug Susceptibility Testing; PZA: Pyrazinamide; MGIT: Mycobacteria Growth Indicator Tube; INH: Isoniazid; S: Susceptible; R: Resistant. Assoc w RI: Associated with resistance–interim.

## Discussion

In this study, we present for the first time, the prevalence of pyrazinamide resistance among rifampicin discordant *M. tuberculosis* isolates. We found a 7.5% prevalence of pyrazinamide resistance which is much lower than the reported resistance of 31 to 89% in MDR-TB [23]. Pyrazinamide resistance was associated with isoniazid resistance and no pyrazinamide resistance was found among drug susceptible TB isolates.

The prevalence of pyrazinamide resistance is known to increase with increasing resistance against other TB drugs [23–26]. A systemic review with meta-analysis reported a median prevalence of pyrazinamide resistance of 51% (31% to 89%) in multidrug-resistant TB [23]. In a study conducted by Allana *et al* in the KZN province, *pncA* polymorphisms was found in 68% of MDR-TB and 96% of XDR-TB isolates [25]. Additionally, in a TB drug resistance survey conducted in South Africa between 2012–2014, the prevalence of pyrazinamide resistance among MDR-TB cases was 59·1% while it was 13.9 among rifampicin mono-resistant cases [26].

In contrast to drug resistant TB, pyrazinamide resistance in drug susceptible TB is reportedly low, ranging from 0 to 10% [23,26,27]. In the South African drug resistance survey, the overall pyrazinamide resistance was 3.7% [26]. With a resistance rate of 7.5%, our study shows that pyrazinamide resistance among discordant isolates was more in keeping with that of drug susceptible TB. Furthermore, when we only considered those isolates that remained discordant at the new WHO reduced rifampicin critical concentration of 0.5 μg/mL, pyrazinamide resistance was even much lower at 2.8%. This underscores the important role of this drug in the treatment of patients with TB disease caused by rifampicin discordant *M. tuberculosis.*

The appropriate diagnosis and treatment of rifampicin discordant TB has not been established. As a result, the WHO recommends standard treatment for RR/MDR-TB until new evidence emerges. Although small retrospective observational studies have reported unfavourable treatment outcomes with standard first line therapy, these studies involved patients who were already resistant to other first line drugs especially isoniazid [2,7]. In contrast, other small studies have reported favourable response to first line therapy in patients who were still susceptible to isoniazid [3,28]. In our study, isoniazid susceptibility was associated with pyrazinamide susceptibility. However, this finding still need to be confirmed in larger multicenter studies. Previous studies have shown that rifampicin discordant *M. tuberculosis* isolates may remain susceptible to rifabutin [11,12,29]. In a recent South African study as many as 33.2% of *rpoB* polymorphisms, were found to be susceptible to rifabutin, the majority of which belonged to the rifampicin discordant mutations [29]. Given the high susceptibility to pyrazinamide shown in this study, as well as the high susceptibility of these strains to rifabutin and isoniazid published elsewhere, it would seem that a significant proportion of patients with rifampicin discordant TB could benefit from rifabutin based first line therapy [11,12]. There is therefore a need for clinical studies investigating the use of such treatment combinations.

Both genotypic and phenotypic pyrazinamide susceptibility testing have limitations. The MGIT method may overcall resistance due to an inoculum effect. A study by Zhang *et al* showed that high inoculum sizes caused an increase in pH over time which leads to loss of acid inhibition, allowing growth of organisms in the presence of the drug [17]. It was however reassuring that all the susceptible isolates in our study were susceptible by the phenotypic method and no *pncA* mutations were detected. Nonetheless, due to the challenges with phenotypic DST, the WHO now recommends genotypic DST for detection of pyrazinamide resistance [30].

Developing a rapid molecular test for pyrazinamide DST has proven difficult as published literature shows lack of hot spot region for *pncA* gene mutations [25]. Our study findings also

confirm this, with four unique mutations and one deletion detected covering a wide range of the *pncA* gene. These mutations are reported to confer resistance to pyrazinamide [22,31]. The WHO recently endorsed the GenoScholar PZA-TB test (Nipro, Osaka, Japan) which is a reverse hybridization molecular assay for detection of pyrazinamide resistance [32]. This assay has a large number of hybridization probes which cover the whole *pncA* gene, making it more complex than the currently used reverse hybridization molecular assays. Implementation of this assay may prove to be more challenging high TB endemic regions. Nevertheless, where pyrazinamide susceptibility testing is not feasible, our study shows that pyrazinamide is most likely to be susceptible in rifampicin discordant TB, even more so if isoniazid susceptible as no pyrazinamide resistance was found in these patients.

This study was not without any limitations. We used manual MGIT for incubation and reading of pyrazinamide phenotypic DST instead of the preferred automated method. This highlights the gap in susceptibility testing of pyrazinamide as this test is not offered in many high TB burdened countries including our laboratory. Because we used both phenotypic and genotypic methods to determine the prevalence of pyrazinamide resistance, we were able to mitigate this limitation. Even the one isolate that failed phenotypic DST had a known pyrazinamide resistance conferring mutation [22,31].

## Conclusion

The current routinely used rapid molecular tests do not differentiate between high level rifampicin resistant mutations and rifampicin discordant mutations. Although the latter constitutes a smaller fraction of the overall DR-TB, the fact that it has unique *rpoB* mutations, and has high susceptibility to pyrazinamide, isoniazid and rifabutin suggests that they may require an alternative first line based treatment regimen. However, a clearer understanding of their clinical significance and their relative contribution to clinical outcomes remains a prerequisite to finding appropriate treatment for these patients.

## Acknowledgments

The authors would like to acknowledge Partson Tinawo for performing statistical analysis for this study.

## Author Contributions

**Conceptualization:** Nomonde Ritta Mvelase.

**Formal analysis:** Nomonde Ritta Mvelase, Ravesh Singh.

**Funding acquisition:** Nomonde Ritta Mvelase.

**Investigation:** Nomonde Ritta Mvelase.

**Methodology:** Nomonde Ritta Mvelase, Ravesh Singh.

**Project administration:** Nomonde Ritta Mvelase.

**Supervision:** Koleka Patience Mlisana.

**Writing – original draft:** Nomonde Ritta Mvelase.

**Writing – review & editing:** Khine Swe Swe-Han, Koleka Patience Mlisana.

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
