## [Decision Letter · Decision Letter 0]

21 Jul 2022

PONE-D-22-16671Pyrazinamide resistance in rifampicin discordant tuberculosisPLOS ONE

Dear Dr. Mvelase,

Thank you for submitting your manuscript to PLOS ONE. After careful consideration, we feel that it has merit but does not fully meet PLOS ONE’s publication criteria as it currently stands. Therefore, we invite you to submit a revised version of the manuscript that addresses the points raised during the review process.

Please submit your revised manuscript. If you will need significantly more time to complete your revisions, please reply to this message or contact the journal office at plosone@plos.org. Please include the following items when submitting your revised manuscript:A rebuttal letter that responds to each point raised by the academic editor and reviewer(s). You should upload this letter as a separate file labeled 'Response to Reviewers'.A marked-up copy of your manuscript that highlights changes made to the original version. You should upload this as a separate file labeled 'Revised Manuscript with Track Changes'.An unmarked version of your revised paper without tracked changes. You should upload this as a separate file labeled 'Manuscript'.

We look forward to receiving your revised manuscript.

Kind regards,

Frederick Quinn

Academic Editor

PLOS ONE

Journal Requirements:

Reviewers' comments:

Reviewer's Responses to Questions

**Comments to the Author**

1. Is the manuscript technically sound, and do the data support the conclusions?

Reviewer #1: Yes

2. Has the statistical analysis been performed appropriately and rigorously? 

Reviewer #1: Yes

3. Have the authors made all data underlying the findings in their manuscript fully available?

Reviewer #1: Yes

4. Is the manuscript presented in an intelligible fashion and written in standard English?

Reviewer #1: Yes

5. Review Comments to the Author

Reviewer #1: GENERAL

Highly relevant contribution to the field of MDRTB. Recent guidelines on the treatment of MDRTB (WHO, 2020) recommended that pyrazinamide only be considered for inclusion in drug regimens if susceptibility to the drug has been shown. Pyrazinamide is a highly effective anti-TB agent and its use would be preferred. The hypothesis in this paper is that isolates resistant to rifampicin by genotypic assay but susceptible to the drug by phenotypic assay (discordant) have a high probability for susceptibility to pyrazinamide (and also to isoniazid). Therefore, the baseline rifampicin profile has major implication for selecting an appropriate treatment regimen that also includes pyrazinamide, especially where DST for the latter agent is not available. It is, therefore, useful to define categories of MDRTB patients where pyrazinamide resistance is likely to be low and its use might be recommended as standard of care.

MAJOR COMMENT

The study investigated pyrazinamide resistance in 80 isolates classified as discordant on rifampicin gDST-pDST results. In these isolates, the pDST of rifampicin was determined using 7H10 medium and a rifampicin critical concentration of 1 ug/ml. In a 2021 technical report on critical concentrations for drug susceptibility testing of isoniazid and the rifamycins (rifampicin, rifabutin and rifapentine), the World Health Organization recommended lowering the CC of rifampicin in 7H10 and MGIT grown cultures to 0.5 ug/ml.

The authors need to elaborate on the implications of this change, if applied to their dataset, on the rationale for the study as stated in the last paragraph of the Introduction. How many of the 80 isolates would still be considered discordant if the revised CC is applied? Do the conclusions for the study hold for isolates regarded as discordant under the revised breakpoint, i.e would the probabilities for association with isoniazid and/or pyrazinamide susceptibility remain unaltered? Are there any high-confidence rpoB mutations in discordant isolates by the revised definition that would predict pyrazinamide susceptibility?

The above additional perspective is to be provided as an expansion of the existing narrative, not necessarily as a replacement of the study results and conclusions as presented.

MINOR COMMENT

The prevalence of pyrazinamide resistance in MDRTB isolates in South Africa is high, as emphasised by the authors. Reference 24 (Ismail, et al. 2018) is quoted in the context of prevalence of rpoB mutations in a national survey. A general perspective on the magnitude of pyrazinamide resistance would be useful in stating the importance of the current work. Does the same reference provide an estimate of the level of pyrazinamide resistance? By comparison, the 2008 paper by Mphahlele, et al., DOI: 10.1128/JCM.00973-08, would be useful, also for insight into the genomics of MDRTB isolates at the time.

6. PLOS authors have the option to publish the peer review history of their article (what does this mean?). If published, this will include your full peer review and any attached files.

Reviewer #1: No

---

## [Author Response · Author response to Decision Letter 0]

11 Aug 2022

Thank you for the opportunity to submit our revised manuscript for further consideration. We appreciate the insightful comments given by the reviewer and we believe that the revised manuscript is of a much better quality. We have addressed each of the comments in the rebuttal letter plus we have also incorporated changes recommended by the reviewer into the manuscript. 

Response to reviewer 1 comments:

MAJOR COMMENT

The study investigated pyrazinamide resistance in 80 isolates classified as discordant on rifampicin gDST-pDST results. In these isolates, the pDST of rifampicin was determined using 7H10 medium and a rifampicin critical concentration of 1 ug/ml. In a 2021 technical report on critical concentrations for drug susceptibility testing of isoniazid and the rifamycins (rifampicin, rifabutin and rifapentine), the World Health Organization recommended lowering the CC of rifampicin in 7H10 and MGIT grown cultures to 0.5 ug/ml.

The authors need to elaborate on the implications of this change, if applied to their dataset, on the rationale for the study as stated in the last paragraph of the Introduction. How many of the 80 isolates would still be considered discordant if the revised CC is applied? Do the conclusions for the study hold for isolates regarded as discordant under the revised breakpoint, i.e would the probabilities for association with isoniazid and/or pyrazinamide susceptibility remain unaltered? Are there any high-confidence rpoB mutations in discordant isolates by the revised definition that would predict pyrazinamide susceptibility?

The above additional perspective is to be provided as an expansion of the existing narrative, not necessarily as a replacement of the study results and conclusions as presented.

Response: Thank you for this comment. The WHO lowering of rifampicin critical concentration (CC) actually strengthens our findings of low pyrazinamide prevalence in rifampicin discordant TB. 45% of the discordant isolates remain discordant after the revision of the CC. Of the six pyrazinamide resistant isolates, only one occurred within this group of isolates, the other five are amongst those who would be considered rifampicin resistant (Rifampicin MIC of >0.5 µg/mL. This means pyrazinamide resistance is even lower in rifampicin discordant TB than originally thought. We have included this information in the description of our M. tuberculosis isolates under the methods section as well as the results section. Additionally, we have expanded our discussion to include the effect of the change on our findings. The specific rifampicin mutations were not helpful in predicting pyrazinamide resistance, probably due to small number of PZA resistant isolates.

MINOR COMMENT

The prevalence of pyrazinamide resistance in MDRTB isolates in South Africa is high, as emphasised by the authors. Reference 24 (Ismail, et al. 2018) is quoted in the context of prevalence of rpoB mutations in a national survey. A general perspective on the magnitude of pyrazinamide resistance would be useful in stating the importance of the current work. Does the same reference provide an estimate of the level of pyrazinamide resistance? By comparison, the 2008 paper by Mphahlele, et al., DOI: 10.1128/JCM.00973-08, would be useful, also for insight into the genomics of MDRTB isolates at the time.

Response: We have also expanded our discussion of pyrazinamide resistance to include this information (paragraph 3 on discussion). Although pyrazinamide resistance is high among drug resistant TB isolates in South Africa, it is much lower in drug susceptible TB. In Ismail’s paper, the overall pyrazinamide resistance was only 3.7%.

---

## [Decision Letter · Decision Letter 1]

2 Sep 2022

Pyrazinamide resistance in rifampicin discordant tuberculosis

PONE-D-22-16671R1

Dear Dr. Mvelase,

We’re pleased to inform you that your manuscript has been judged scientifically suitable for publication and will be formally accepted for publication once it meets all outstanding technical requirements.

Kind regards,

Frederick Quinn

Academic Editor

PLOS ONE

Additional Editor Comments (optional):

Reviewers' comments:

Reviewer's Responses to Questions

**Comments to the Author**

1. If the authors have adequately addressed your comments raised in a previous round of review and you feel that this manuscript is now acceptable for publication, you may indicate that here to bypass the “Comments to the Author” section, enter your conflict of interest statement in the “Confidential to Editor” section, and submit your "Accept" recommendation.

Reviewer #1: All comments have been addressed

2. Is the manuscript technically sound, and do the data support the conclusions?

Reviewer #1: Yes

3. Has the statistical analysis been performed appropriately and rigorously? 

Reviewer #1: Yes

4. Have the authors made all data underlying the findings in their manuscript fully available?

Reviewer #1: Yes

5. Is the manuscript presented in an intelligible fashion and written in standard English?

Reviewer #1: Yes

6. Review Comments to the Author

Reviewer #1: The authors have responded positively to previous recommendations, especially as far as the Major comment raised is concerned. The manuscript now contains additional considerations of the study outcome in the light of recent changes in international recommendations for the rifampicin breakpoint in liquid media. Minor comments by the reviewer have also been adequately addressed. There are no additional clarifications or amendments required.

7. PLOS authors have the option to publish the peer review history of their article (what does this mean?). If published, this will include your full peer review and any attached files.

Reviewer #1: No

---

## [Editor Report · Acceptance letter]

12 Sep 2022

PONE-D-22-16671R1 

Pyrazinamide resistance in rifampicin discordant tuberculosis 

Dear Dr. Mvelase:

I'm pleased to inform you that your manuscript has been deemed suitable for publication in PLOS ONE. Congratulations! Your manuscript is now with our production department. 

Kind regards, 

on behalf of

Dr. Frederick Quinn 

Academic Editor

PLOS ONE